# Memory-Based Sequential Attention

**Jason Stock**                                                    STOCK@COLOSTATE.EDU

**Charles Anderson**                                        ANDERSON@COLOSTATE.EDU

*Department of Computer Science, Colorado State University*

## Abstract

Computational models of sequential attention often use recurrent neural networks, which may lead to information loss over accumulated glimpses and an inability to dynamically reweigh glimpses at each step. Addressing the former limitation should result in greater performance, while addressing the latter should enable greater interpretability. In this work, we propose a biologically-inspired model of sequential attention for image classification. Specifically, our algorithm contextualizes the history of observed locations from within an image to inform future gaze points, akin to scanpaths in the biological visual system. We achieve this by using a transformer-based memory module coupled with a reinforcement learning-based learning algorithm, improving both task performance and model interpretability. In addition to empirically evaluating our approach on classical vision tasks, we demonstrate the robustness of our algorithm to different initial locations in the image and provide interpretations of sampled locations from within the trajectory.

**Keywords:** sequential attention, transformers, interpretability, reinforcement learning

## 1. Introduction

The human visual system constantly receives a vast amount of data, with estimates ranging from $10^8$ and $10^9$ bits of information per second Borji and Itti (2012); Koch et al. (2006). Given the sheer volume of this input, it is crucial to have some mechanisms in place for filtering out extraneous or erroneous data to effectively process it in real-time. To accomplish this task, the visual system relies on advanced cognitive processes and forms of dynamic attention. Underlying this fundamental principle are evolved mechanisms for selection based on some notion of relevance.

The basis for many computational models of attention build on the pinnacle work of Treisman and Gelade (1980), who proposed the "Feature Integration Theory". Intuitively, this theory suggests that attention can be directed towards visually distinct regions or features that stand out in comparison to their surroundings. The importance, relating to human perception, is that the entire visual scene is not processed at once. Rather, we selectively build an internal representation based on localized information. A given location or memory of previous locations may be informative for where to look next, where the total history of locations may influence scene interpretations. However, in recent years, the primary focus of most neural models is on improving on or learning salient features, or the relationships thereof, rather than on identifying the sequential shifts of attention that are used for inference Borji and Itti (2012); Riche et al. (2013).

In this work, we take inspiration from biological models of visual attention for describing global scene understanding with visual scanpaths or trajectories. We propose a memory-based sequential model of attention[1] leveraging the transformer Vaswani et al. (2017); Dosovitskiy et al. (2020) to contextualize the history of local information to learn "where to look"

---

1. https://github.com/stockeh/memory-sequential-attention

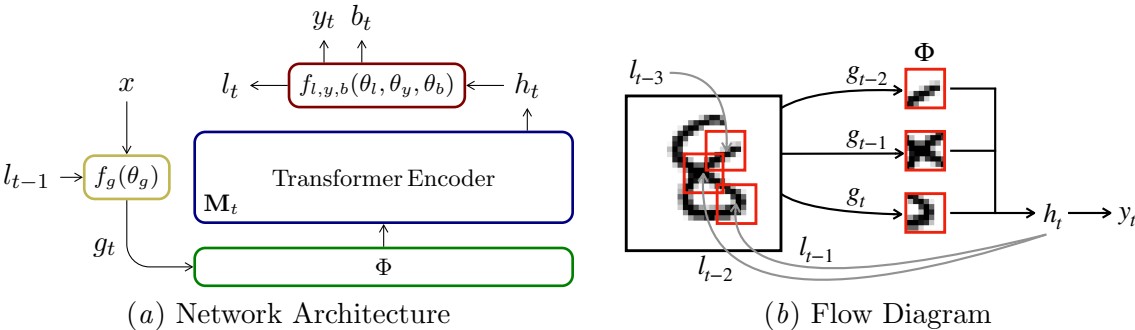

Figure 1: High-level architectural overview of our proposed model. A predicted location $l_{t-1}$ samples a glimpse, $g_t$, from an input image, $x$. The glimpse is added to a memory store, $\Phi$, where a masked transformer block computes a hidden state, $h_t$. The hidden state is used to emit the next location, $l_t$, a predicted class, $y_t$, and baseline estimate, $b_t$.

for classifying a visual scene (Figure 1). A hybrid approach of reinforcement learning and data likelihood estimation is used to find a control strategy with an optimal trajectory over a visual scene. The result is a performant model that promotes interpretability, enabling the assessment of influence from individual locations for a given task. It also enables the interpretations of interactions between these locations from the distribution of self-attention weights. Results are evaluated on classical vision tasks, demonstrating the validity of our approach and shows competitive performance.

The remainder of this work is organized as follows. An overview on the cognitive theories and neural models of attention, as they relate to this work, are provided in Section 2. In Section 3 we detail our methodology and specifics of our proposed architecture. We then evaluate our experimental results in Section 4. In Section 5 we discuss network interpretations and make comparisons with a vision transformer. Lastly, concluding remarks summarizing our work are made in Section 6.

## 2. Background and Related Work

To get a better intuition of computational models of attention, we first review these concepts as a cognitive process, where the fundamental theories on visual control from psychology and neuroscience are introduced. Thereafter, we discuss neural models of attention and how they relate to this work.

### 2.1. Biological Visual Attention

Classical studies in cognitive psychology and neuroscience isolate selective attentional control to follow two predominate theories in vision Theeuwes (2010); Failing and Theeuwes (2018). The first is a *top-down* process (also called endogenous or goal-directed), whereby control is volitional and activated by the observer. This may use one's belief and 'internal' factors to control attention. On the contrary, a *bottom-up* (also called exogenous or

stimulus-driven) theory suggests control is involuntarily driven by factors external to the observer. Stimuli that are physically salient due to their inherent properties relative to the surrounding environment are likely to capture attention. Here, the salience of a stimulus is defined by low-level visual characteristics, including modalities of color, intensity, orientation, and motion Itti and Koch (2001).

Both theories of attentional control are shown to work in tandem for selective attention, but these processes alone do not fully explain the range of phenomena related to attention. Only recently, however, the dichotomy of top-down and bottom-up control has been challenged with evidence for additional factors that control visual selection, such as *reward-based history* effects Failing and Theeuwes (2018); Awh et al. (2012). The underlying idea is that past episodes of attentional selection can strongly influence current selection above and beyond top-down and bottom-up processing. This is particularly evident in studying the interactions between rewards and attention, where rewards can shape both perceptual and attentional processes, prioritizing certain stimuli and modifying spatial and temporal attentional selection.

The processes of top-down, bottom-up, and reward-based history also control both covert and overt processing of visual information Kowler (2011); Failing and Theeuwes (2018); Zhao et al. (2012). *Covert attention* refers to the internal processing of visual information without any *saccadic eye-movements*, the rapid eye-movements that occur when shifting gaze between locations. This allows for parallel processing and quick interpretations of visual information. It is intuitively used to monitor the environment and guide eye-movements, allowing us to attend to a target without fixation.

*Overt attention*, on the other hand, is associated with a fixation as eye-movements direct attention to different locations in the environment or visual scene. Attentional focus, therefore, occurs within the line of sight of the *fovea*, the central part of the retina responsible for sharp, central vision. Overt attention is an intentional and conscious process that allows us to focus on specific stimuli in the scene, often to gather more detailed information related to a given task.

## 2.2. Computational Models of Attention

As early as 1987, Koch and Ullman conceptualized a feed-forward model to aggregate salient features based on color, intensity, and orientation in order to compute a saliency map emphasizing conspicuous locations. A "winner-take-all" approach, based on the concept of inhibition of return Tipper et al. (2003), is then utilized to shift the focus of attention to the next salient region. This approach was later implemented and validated as a computational model for use with digital images Niebur and Koch (1995); Itti et al. (1998); Itti and Koch (2001).

Since these early works in the field, much effort has been devoted to modeling saliency maps and the development of attention for predictive vision tasks. These are often controlled through a combination of top-down and bottom-up processes, and involve internal computations within the model. We define the category of spatial and contextual attention as feed-forward methods. The importance of discussing these methods is as a precursor to motivate sequential attention.

Soft spatial attention methods resolve salient features with a continuous-value mask. This can occur in the spatial domain Song et al. (2022); Woo et al. (2018); Xu et al. (2015) or as a special extension over channels Hu et al. (2018); Woo et al. (2018). In contrast, hard attention localizes and crops selective regions to process for the relevant task. This can be done by learning a transformation over the input Jaderberg et al. (2015), with region proposals Anderson et al. (2018), or by learning a masking strategy Li et al. (2020); Wang et al. (2018).

Contextual attention is inspired by the relationship between top-down and bottom-up visual cues that explain attentional deployment. In this context, the guidance of selection bias over *values* (sensory inputs) with attention pooling considers the interactions of a given *query* (top-down, volitional cue) and a set of *keys* (bottom-up, nonvolitional cues). This relationship is more commonly modeled with scaled dot-product attentional pooling, motivating the transformer architecture Vaswani et al. (2017). A natural extension from natural language to visual scenes was proposed by Dosovitskiy et al. (2020). As a result, attention weights correspond to the contextual relationships of individual locations. However, such locations consider the entire input, which may be irrelevant for the task, and incur unnecessary computations.

Instead of identifying fixation zones through salient, bottom-up or top-down features, some studies consider attention as a sequential decision making process Larochelle and Hinton (2010); Mnih et al. (2014); Ba et al. (2014); Welleck et al. (2017); Elsayed et al. (2019); Kumari and Chakravarthy (2022); Schwinn et al. (2022), as we do here. In certain cases, we can model the reward-based history effects of visual control to direct the learning of trajectories that are task relevant with reinforcement learning. The task of where to attend, therefore, becomes a sequential learning problem with covert sampling of a sensory scene.

Many works build on the foundational work from Mnih et al. (2014), where a recurrent neural network accumulates information over time to decide how to act. The primary transfer of temporal information is combined through the hidden state of the recurrence. As such, the final step class prediction relies on the accumulation of state representations. This can lead to information loss as states are accumulated and prior representations are unable to be dynamically re-weighted.

In this work, we replace the recurrence with a single transformer encoder comprised of multiple self-attention heads. To an extent, this is similar to the work from Dosovitskiy et al. (2020), who introduce the vision transformer using patches that equally span the visual scene. However, our approach samples patches one at a time from continuous locations within the environment. This sampling reduces the overall sequence length and results in features that can be dynamically and contextually attended.

## 3. Memory-Based Sequential Attention

Instead of processing the entire image $x$ at once, we sample smaller patches, or glimpses, for sequential decision making. Our proposed model (Figure 1) stores the memory of previously visited glimpses from a trajectory, and contextualizes their relationships to learn a strategy of "where to look". This approach uses reinforcement learning and data likelihood estimation to optimize for classification tasks.

### 3.1. Preliminaries

The recurrent model of visual attention (RAM) Mnih et al. (2014) relies on a recurrent neural network to sample glimpses. At each time step $t$, the agent observes a glimpse of the environment from a particular continuous-valued location, $l_t = (i_t, j_t)$, and accumulates this over time to determine a location for the next step. A scalar reward is emitted at each step, where in a classification setting, a positive value is given if the class is correctly predicted. The goal is, therefore, to select a sequence of observations from the environment that maximize the total cumulative reward.

In this work, we replace the recurrent network state representation with a modified vision transformer (detailed in Section 3.2). However, we leverage from RAM the following network components:

**Glimpse Network**   The previous location $l_{t-1}$ is used to sample a retina-like representation (or glimpse) $\rho(x, l_{t-1})$ from the full image, $x$, providing the agent only a partial view of the scene at time $t$. An initial location is set as $l_0 = (0, 0)$ or drawn randomly within some range between $[-1, 1]$, where $(-1, -1)$ is the top left and $(1, 1)$ is the bottom right. The resolution of a glimpse is defined by the number of scales, $s$, composing high- and progressively lower-resolution regions around the location, stacked as separate channels.

We use non-linear, fully-connected layers to extract the embedding of a given glimpse and model "what" it represents. We also use the transformed location to capture "where" the glimpse is located. We then combine the output of these two models, capturing the "what-where" combination, through a subsequent non-linear transform to create a glimpse feature vector represented by $g_t = f_g(x, l_{t-1}; \theta_g)$.

**Location Network**   At every timestep, a location, $l_t$, is emitted by the location network, $f_l(h_t; \theta_l)$, using the hidden state, $h_t$, of the model as calculated in Section 3.2. The location is stochastically sampled from a parameterized Gaussian distribution with fixed variance as $l_t \sim p(\cdot | f_l(h_t; \theta_l))$, where the mean of the $i, j$ coordinates are estimated by the location network. In the context of reinforcement learning, we sample values from the location policy, where the policy function maps the current observation of the environment (in this case, the hidden state, $h_t$) to the action to be taken by the agent (the continuous-valued location, $l_t$). During training, the log probabilities of the sampled locations are used to update the network.

**Classification Network**   Similar to the location network, we use the hidden state, $h_t$, to predict a class by passing it through an additional network that outputs the softmax over a set of possible classes. Specifically, we represent this as $y_t = \arg\max_c p(c \,|\, f_y(h_t; \theta_y))$ from which a class is selected.

### 3.2. Contextual Attention over Memory

Discovering the next best location of where to look or interpreting what has been seen over some sequence is a complex task. It is not always the case that every observed location is relevant. However, the relationship between different regions, or even from a single location, may be informative for a given task. This assumes ample exploration of the visual scene with memory of what has already been observed.

Using the glimpses, or multi-resolution patches, that we sample from the visual scene (as described in Section 3.1), we populate a *memory store*, $\Phi = \{g_0, \ldots, g_k\}$, that buffers the history of all observed glimpses in the trajectory. Treating each glimpse as a token, we use self-attention to contextualize their relationship and highlight important locations to compute an embedding, $h_t \in \mathbb{R}^d$, that is used to emit the next location or class prediction. A traditional vision transformer Dosovitskiy et al. (2020) will partition an image, $x \in \mathbb{R}^{c \times h \times w}$, into $n$ equally divisible patches of size $p$ such that $n = (w/p) \cdot (h/p)$ with dimension $d = p^2 c$. However, $\Phi$ has a sequence length $k \ll n$ from sequentially concatenated glimpses with an embedding dimension $d$ from the glimpse network.

The set of tokens in memory have a standard sine-cosine positional encoding added with a padding of zero-valued vectors to have a fixed length $k$. Let matrix $\mathbf{X} \in \mathbb{R}^{k \times d}$ be the new row-wise concatenation of the tokens. We begin to compute $h_t$ with a single transformer block composed of multi-head self-attention (MSA) and a residual point-wise fully-connected network (FCN) as,

$$\mathbf{Z} = \text{FCN}(\text{MSA}(\mathbf{X})) \quad \text{such that} \tag{1}$$

$$\text{MSA}(\mathbf{X}) = [\mathbf{O}_1, \mathbf{O}_2, \ldots, \mathbf{O}_h]\mathbf{W^O}, \tag{2}$$

where $h$ is the number of heads, $\mathbf{W^O} \in \mathbb{R}^{hv \times d}$ are trainable weights, $[\cdot]$ is the column-wise concatenation, and $\mathbf{O}_i \in \mathbb{R}^{k \times v}$ is the output of the $i$-th attention head with latent dimension $v < d$. We introduce a mask $\mathbf{M} \in \mathbb{R}^{k \times k}$ to ignore the padded and yet to be observed locations and compute each head as,

$$\mathbf{O}_i = \mathbf{A}_i \mathbf{V}_i \quad \text{such that} \tag{3}$$

$$\mathbf{A}_i = \text{softmax}\big((\mathbf{Q}_i \mathbf{K}_i^\top + \mathbf{M})/\sqrt{d}\big) \in \mathbb{R}^{k \times k}. \tag{4}$$

This mask effectively pushes the attention weights of padded locations, across all batches, in the softmax toward zero with,

$$\mathbf{M}_{*,j} = \begin{cases} -\infty & \text{if } j \geq t \\ 0 & \text{otherwise} \end{cases} \tag{5}$$

The queries, $\mathbf{Q}_i$, keys, $\mathbf{K}_i$, and values, $\mathbf{V}_i$ are found via a linear projection of $\mathbf{X}$ by,

$$\mathbf{Q}_i = \mathbf{X}\mathbf{W}_i^Q, \quad \mathbf{K}_i = \mathbf{X}\mathbf{W}_i^K, \quad \mathbf{V}_i = \mathbf{X}\mathbf{W}_i^V, \tag{6}$$

with trainable weight matrices $\mathbf{W}_i^Q, \mathbf{W}_i^K, \mathbf{W}_i^V \in \mathbb{R}^{d \times v}$.

The FCN is a two layer residual network separated by the ReLU activation, $\delta$, and dropout ($p = 0.2$) that takes as input the layer normalized (LN) residual output from above, defined by $\bar{\mathbf{X}} = \text{LN}(\mathbf{X} + \text{MSA}(\mathbf{X}))$. We then compute this as,

$$\text{FCN}(\bar{\mathbf{X}}) = \text{LN}(\bar{\mathbf{X}} + \delta(\bar{\mathbf{X}}\mathbf{W}^R)\mathbf{W}^S), \tag{7}$$

where $\mathbf{W}^R \in \mathbb{R}^{d \times m}$ and $\mathbf{W}^S \in \mathbb{R}^{m \times d}$ such that $m > d$. The output is reshaped from $k \times d \rightarrow kd$ and linearly projected into $h_t = \mathbf{Z}\mathbf{W^z}$ with $\mathbf{W^z} \in \mathbb{R}^{dk \times d}$ (see Equation (1)). This hidden state is used for the subsequent network components and repeats at every timestep with an updated $\Phi$ until termination.

Note that the effective padding and masking steps could be left without (in Equation (4)) by allowing $\Phi$ to have a dynamic sequence length. The output of scaled-dot product attention will also have a variable length. Thus, we can compute the mean over the sequences to obtain a $d$-dimensional vector, yielding $h_t$. However, we find this approach to be insufficient as the re-weighted glimpse extrema are important indicators for $h_t$.

### 3.3. Training Procedure

We view the problem of "where to look" as a control problem, or Partially Observed Markov Decision Process (POMDP), where the next transition only depends on the current state (*i.e.*, memory of previous glimpses) and action (*i.e.*, continuous-valued location). The objective, as a reinforcement learning problem, is to learn a strategy or sequence of actions that maximizes the cumulative reinforcements along a trajectory. At each timestep an agent selects an action, $a_t \in \mathcal{A}$, in the current state, $s_t$, according to its policy. A scalar reward $r(s_t, a_t)$ is received and then transitions to the next state $s_{t+1}$ following the probability $s_{t+1} \sim P(\cdot|s_t, a_t)$.

Consider a stochastic policy $\pi_\theta$, parameterized by a neural network, such that we aim to maximize the expected return $J(\pi_\theta) = \mathbb{E}_{\tau \sim \pi_\theta}[R(\tau)]$. We assume an episodic environment with $\tau = (s_0, a_0, \ldots, s_{T+1})$ where we can estimate the expectation with a sample mean given a set of trajectories, $\mathcal{D} = \{\tau_i\}_{i=1,\ldots,N}$. By the policy gradient theorem, and shown by Williams (1992), we arrive at an approximation to derive the analytical gradient,

$$\nabla_\theta J(\pi_\theta) \approx \frac{1}{|\mathcal{D}|} \sum_{\tau \in \mathcal{D}} \sum_{t=0}^{T} \nabla_\theta \log \pi_\theta(a_t|s_t) \left(R(\tau)_t - b(s_t)\right). \tag{8}$$

In this approximation, we include a baseline that does not depend on the action, *e.g.*, an estimate of the value function $b(s_t) = \mathbb{E}_\pi[R(\tau)_t]$, to reduce variance and improve convergence. We emit this baseline along with the next location $l_t$. The gradient of this expected return or learning rule (Equation (8)) is also referred to as *REINFORCE with baseline*

Following Mnih et al. (2014), only the location network is trained by maximizing $J(\pi_\theta)$. In doing so, the gradient information does not flow to any other network components. We train a baseline network, for use with optimizing the locations, by minimizing the mean squared error with the rewards at each step, $\mathcal{L}_b = \sum_{t=0}^{T}(b_t - r_t)^2$, again, restricting gradient flow to the rest of the model. All other network components are updated to minimize the cross entropy, $\mathcal{L}_y = -\sum_{c=1}^{M} t_{i,c} \log(p_{i,c})$, with ground truth class labels.

## 4. Experiments

We evaluate our method for classification on MNIST and cluttered MNIST datasets. Using a predefined number of glimpses, we output our final class prediction at the last step. During training, a reward of 1 is assigned if the target class is correctly predicted, and 0 otherwise, assigning all previous steps this value. Results are compared to differing network architectures of increased complexity. As a baseline we include a fully-connected and convolutional network with ReLU non-linearities where we define the number of units and filters in Table 1. We also compare results with a standard vision transformer Dosovitskiy

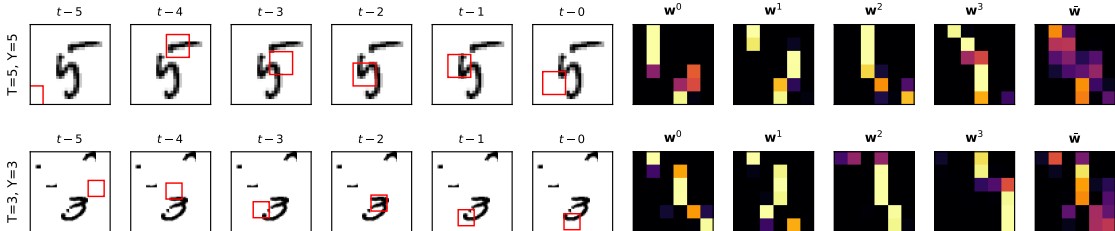

Figure 2: The learned policy of our best models and distribution of attention weights. (top) An example from MNIST and (bottom) from cluttered MNIST dataset. Columns 1-6 illustrate the trajectory of individual glimpses in red from the first timestep on the left up to the point of prediction. Columns 7-11 visualize the distribution of self-attention weights for each attention head with the associated mean in the last column.

et al. (2020), which has a fixed patch size that equally subdivides an image over the entire input space.

Our comparison and implementation of RAM follows the architecture as described by Mnih et al. (2014). However, as there were no reported hyperparameters, our results slightly differ from the original. In our approach, we maintain hyperparameter consistency with replacement of the recurrent network for our memory-based transformer. A complete list of parameter values (*e.g.*, optimizer, training epochs, etc.) are saved and can be found in our code repository. In the following experiments we use a single transformer block and vary the number of self-attention heads between 1 and 4.

All models are implemented in PyTorch with experiments conducted on a single node with an NVIDIA GeForce RTX 3090 (24GB), Intel i9-11900F (2.50GHz), and 128GB memory. All models have a similar number of trainable parameters.

## 4.1. MNIST Classification

We use the MNIST dataset, of size $28 \times 28$, to demonstrate the effectiveness of our approach. Data are partitioned into training (50000), validation (10000), and test (10000). The validation data is only used for hyperparameter tuning. Evaluation results are summarized in Table 1(*a*), where the top-1 classification error is reported on the test set. We use a sequence length of $k = 6$ with a glimpse size of $8 \times 8$ and a scale $s = 1$ for our approach and with RAM.

As we increase the number of self-attention heads from 1 to 4, we notice a steady decrease in error that eventually plateaus around 1.00%. However, we find that our performance is competitive to the implementation of RAM. Interestingly, we achieve a lower error than that reported in their original paper (cf. Mnih et al. (2014), Table 1[2]). We speculate this is as a result of hyperparameter tuning.

The vision transformer uses a patch size of $7 \times 7$ that equally spans the image domain. The number of heads are equal to our best performing model with the same linear transform

---

2. Officially reported top-1 classification error of 1.07%.

Table 1: Classification results where S is the glimpse scale, H is the number of attention heads, and K is the number of sequential glimpses.

(a) MNIST

| MODEL | ERROR |
|---|---|
| FC, 2 LAYERS [256, 256] | 2.20 |
| CNN, 2 LAYERS [16, 32] | 1.17 |
| VIT, 7 × 7, 4H | 3.48 |
| RAM, 6 K, 8 × 8, 1 S | **0.94** |
| OURS, 6 K, 8 × 8, 1 S, 1 H | 1.29 |
| OURS, 6 K, 8 × 8, 1 S, 2 H | 1.11 |
| OURS, 6 K, 8 × 8, 1 S, 4 H | **1.05** |

(b) Cluttered & Translated MNIST

| MODEL | ERROR |
|---|---|
| FC, 2 LAYERS [256, 256] | 56.82 |
| CNN, 4 LAYERS [8, 16, 32, 64] | 6.71 |
| VIT, 12 × 12, 4H | 29.48 |
| RAM, 6 K, 12 × 12, 3 S | 6.43 |
| OURS, 6 K, 12 × 12, 3 S, 1 H | 7.89 |
| OURS, 6 K, 12 × 12, 3 S, 2 H | 7.47 |
| OURS, 6 K, 12 × 12, 3 S, 4 H | **6.20** |

Table 2: Policy error (mean ± std) when permuting the starting location in our best model, then following the learned policy. The last line is a stochastic policy that samples a random action (location) at each step.

| POSITION | MNIST | CLUTTERED |
|---|---|---|
| RANDOM | **1.12 ± 0.03** | 6.76 ± 0.07 |
| TOP-MIDDLE | 1.13 ± 0.05 | 7.13 ± 0.11 |
| TOP-LEFT | 1.16 ± 0.03 | 7.02 ± 0.22 |
| CENTER | 1.20 ± 0.05 | **6.36 ± 0.19** |
| BOTTOM-MIDDLE | 1.20 ± 0.07 | 7.32 ± 0.15 |
| BOTTOM-RIGHT | 1.12 ± 0.05 | 6.74 ± 0.18 |
| RANDOM POLICY | 29.49 ± 0.28 | 25.66 ± 0.03 |

and feed-forward embedding dimensions. We find the performance is worse than all of our tested models. This suggests that our learned policy is able to identify the most task-relevant glimpses from a shorter sequence length.

The top row of Figure 2 shows a sample from our best performing model with a trajectory learned by the policy. A red box surrounds the glimpse locations from each timestep as it moves throughout the image. The model only observes the cropped information inside of this box and all outside information is discarded. To the right are the distribution of attention weights for each head with their associated mean. Attention weights are associated with the most conspicuous location. We find locations over the digit with the highest attention weights, whereas uninformative locations have low weight. This result validates our intuition as to what locations represent a digit. Additional details for interpreting these weights are made in Section 5.1, with more examples in Appendix A.

**4.2. Cluttered and Translated MNIST**

To further test our approach we evaluate results on the cluttered MNIST dataset Fidjeland (2015). This data contains an MNIST digit that is randomly translated within a $60 \times 60$ canvas. Four different $8 \times 8$ subpatches sampled from other random digits are added at random locations. The presence of clutter as a form of noise make the task particularly challenging. Compared to the centered MNIST data, accurate predictions are more dependent on a model that is invariant to translation and can learn to ignore the clutter that is not task-relevant. Data are partitioned into training (50000), validation (10000), and test (10000).

Table 1($b$) shows the classification results from the different architectures. For our model and our RAM implementation, we use a glimpse size of $12 \times 12$ with $s = 3$ to capture multi-resolution features over a sequence length of $k = 6$. Similarly, the vision transformer uses the same patch size with 4 self-attention heads. The benefits of our model is especially noticed when comparing to the vision transformer. RAM and our model achieve less than 7% error, while the vision transformer achieves about 29.5% error with the same number of attention heads.

Results show the fully-connected network performs worst, whereas the convolutional network performs significantly better with its inductive biases of translation invariance. However, our memory-based attention model shows a slight advantage as we learn a policy to avoid the clutter and focus on the different parts of the digit. We outperform RAM with the advantage of having attention weights that bring insight to how these glimpses from memory relate to each other.

As with the MNIST digits, we show an example trajectory and our model's attention weights in the bottom row of Figure 2. We find that for the last step class prediction, the glimpses from memory attend primarily to those that are directly focused on the digit, illustrating how our policy avoids the clutter while exploring the visual scene. Alternatively, indices with near zero values are found where the high-resolution glimpse is not focused directly on the digit.

**4.3. Location Permutations**

To evaluate model robustness we make inference with our best performing models and compare the results to a random policy. The results for each dataset are shown in Table 2, where the random policy has locations sampled from a uniform distribution at each step. Each of the six starting positions tested herein follow the learned policy, but vary in where the first glimpse is sampled. This effectively shows the learned policy, with $\sim 20\%$ lower error, optimally selects locations for the given task.

There are slight variations in classification accuracy for different starting locations. The change is minimal with MNIST, but with the cluttered and translated MNIST dataset, there is a noticeable influence. Namely, we find lower error when the initial glimpse is sampled from the center of the image, and a higher error when sampling from the periphery. We speculate this to be as a result of there being greater coverage of the multi-resolution glimpse, with periphery information, that can more accurately resolves the high resolution features in subsequent steps.

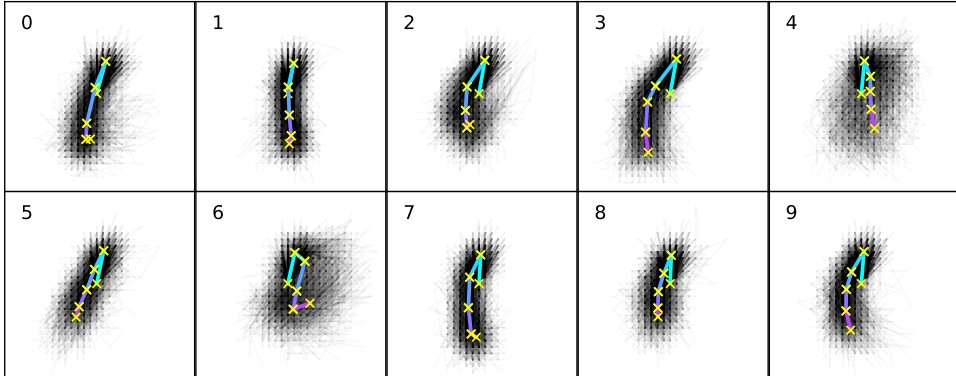

Figure 3: Class specific trajectories of all MNIST test samples in black with the mean trajectory overlaid. The mean glimpse locations, ×, all have a centered initial position, then progress following the path from cyan to purple.

In Figure 3, we show how inference trajectories vary among MNIST test samples for each class independently. By taking the trajectory mean, across all samples in each class, we glean insights into the global path and policy they follow. The following observations are based on this mean. Generally, for each class, the second glimpse is made at the top of each digit with a trend, scanning down the image that follows. The differences between each class are evident and supported by the structure of the digit. Take, for example, class '6', where the final glimpse moves right to sample the commonly enclosed circle of the digit. Without this, the sampled glimpses are similar to a '1'. Class '3' trajectories scan further left, presumably to delineate between class '8'. Lastly, class '4' trajectories are further spread around the top of the digit.

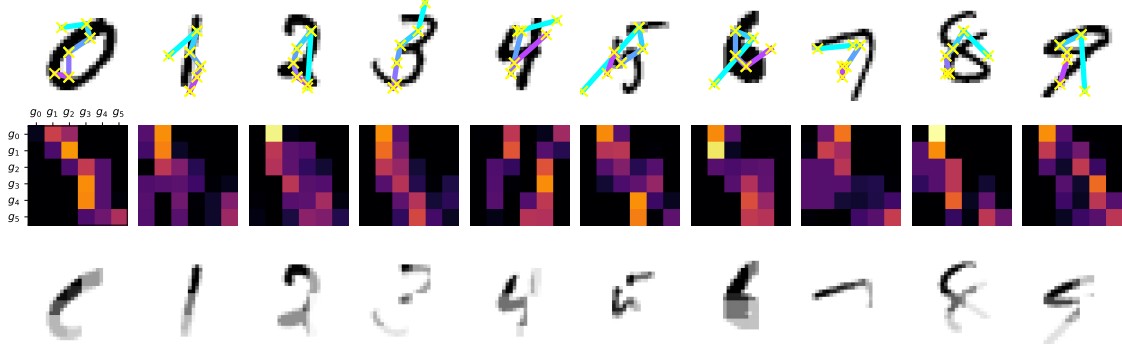

Figure 4: (top) Sampled glimpse locations, ×, from within a trajectory, starting in cyan and ending in purple. (middle) Mean self-attention weights computed over all heads for glimpses $g_{0-5}$, where yellow is 1 and black is 0. (bottom) Re-weighted glimpse locations found by the accumulated attention weights.

## 5. Discussion

In this section we discuss how our approach improves network interpretability/transparency by reasoning over predictions, and then we make comparisons with the vision transformer.

### 5.1. Network Interpretations

Learning a strategy of where to look results in an interpretable process of decision making, allowing for the assessment of influence from individual glimpses for a given task. Manual review of these locations can help one reason about how the network arrives at the prediction. This is as a result of reducing the problem to a subset of locations from the entire scene, ignoring extraneous or erroneous data. However, it is also important to consider how the network uses these locations beyond human intuition. In our approach, we provide additional transparency by contextualizing over the observed glimpses in memory. By inspecting the distribution of self-attention weights we can glean insights to how these locations are attended to.

Figure 4 shows the final step trajectory for different MNIST digits. The mean over attention heads is computed to capture comprehensive relevance. Along each axis are weights corresponding to the relationship of each glimpse, such that the diagonal represents how a glimpse attends to itself. Interestingly, we find that glimpse locations that are task irrelevant, *i.e.*, zero-valued locations or at locations with clutter, have little to no positional significance in the sequence. By contrast, the locations centered on the most conspicuous locations of the digit are largely attended to.

To exemplify this intuition, we accumulate the mean attention weights of each column to weigh each individual glimpse location. The last row of Figure 4 shows this result, where the target regions of increased clarity correspond to the most attended locations in the trajectory. Intuitively, the re-weighting of glimpse locations make sense as they show the features that are most unique to a given digit.

Additional insights on the impact of subsequent glimpses are made by emitting the class prediction after every timestep. This is helpful to understand more generally how the inclusion of each glimpse contribute to the improvement in model performance. Figure 5 illustrates this result for our model as it compares to RAM on both datasets. With MNIST (Figure 5($a$)), we find a 31% and 19% average increase in accuracy after glimpses $g_1$ and $g_2$ are observed, respectively. For the cluttered dataset (Figure 5($b$)), we find a 20% increase in accuracy in the first glimpse and marginally higher accuracy following. These findings indicate that our location policy can quickly identify locations that our model can contextualize over for classification. Furthermore, displaying the general and steady increase in performance that eventually plateaus with more glimpses.

### 5.2. Comparison to a Vision Transformer

Self-attention has a complexity of $\mathcal{O}(n^2 \cdot d)$ that is quadratic in the sequence length, $n$. Standard vision transformers assume the image size is divisible by the patch size, $p$, to compute the constant sequence length. The computational and memory cost, therefore, is especially noticeable for a large image, as $p$ decreases, and as the number of attention layers increase. Pruning irrelevant tokens within hidden layers, *e.g.*, with sparse transformers that

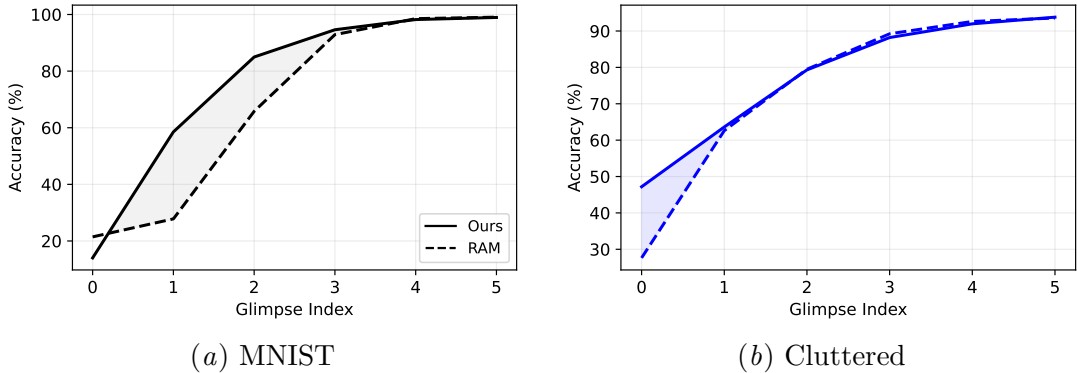

Figure 5: Test accuracy emitted after each glimpse of our approach as it compares to RAM. The shaded region emphasizes the performance improvements of early glimpses

use additional prediction networks Rao et al. (2021); Meng et al. (2022) or scoring functions Xu et al. (2022); Yin et al. (2022); Liang et al. (2022), is one such way to reduce complexity. However, these approaches still operate on the full-sized image as input and the non-linear combination of tokens over layers makes it challenging to interpret their true representation.

In this work, we learn a control strategy to sequentially sample $k \ll n$ ideal patch locations from the input, *i.e.*, selecting relevant patches rather than removing them. This improves interpretability of relevant tokens and more closely aligns our model to the biological visual system. Furthermore, we do not need to assume the number of patches is a product of the image size, thus, allowing our method to scale to any size input.

Additionally, vision transformers do not have any inductive biases of translation or scale invariance. This means that objects that have a relative change in their position will result in a different response. While these aspects can be learned with ample data or through augmentations, it can still hinder performance. This is evident when comparing a convolutional network, that is translation invariant, to the vision transformer on cluttered MNIST (Table 1(*b*)). By contrast, our location policy samples a glimpse along a trajectory that is optimized for the task. This inherently results in features that are invariant to translation by learning to ignore irrelevant locations. As such, our method benefits from the properties of vision transformer models while also being translation invariant with a shorter sequence length.

## 6. Conclusion

In this work we introduce a memory-based sequential attention model that combines information over a subset of image locations for classification. Our proposed memory module incorporates a transformer architecture, enabling contextualization of relationships among sequential glimpses. This leads to the development of a more interpretable policy for selecting regions, optimized through reinforcement learning to maximize data likelihood. Consequently, our approach yields improved network transparency by sequentially highlighting informative region proposals.

Our experimental results demonstrate that our model outperforms baseline architectures in classical vision tasks. Although we do not achieve state-of-the-art performance, we consider this work a significant step toward enhancing model interpretability, aligning more closely with the principles of the biological visual system. This is especially pertinent to bridge the gap between machine learning and practical applications domains such as medical diagnoses and modeling weather and climate. In future work, we plan to apply our model to data within these domains. Furthermore, we intend to explore the use of saliency measures to guide initial glimpse locations, further reducing the overall sequence length. Lastly, we would like to disentangle whether glimpse weights are most important for the next location or output prediction by evaluating gradient based sensitivity measures.

## Acknowledgments

This work is supported by NSF Grant No. 2019758, *AI Institute for Research on Trustworthy AI in Weather, Climate, and Coastal Oceanography (AI2ES)*.

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

## Appendix A. Glimpse Trajectories

Here we show visualizations of the learned policy for our best models and distribution of attention weights with examples from each class in the MNIST (Figure 6) and cluttered MNIST (Figure 7) test datasets. Columns 1-6 illustrate the trajectory of individual glimpses, which are cropped from the original image, in red from the first timestep on the left up to the point of prediction. The class prediction $y$ is labeled next to the target image $t$. Note that some samples in Figure 7 are incorrectly classified, and evidently have a poor trajectory. Columns 7-11 visualize the distribution of self-attention weights from the 4 attention heads and their associated mean.

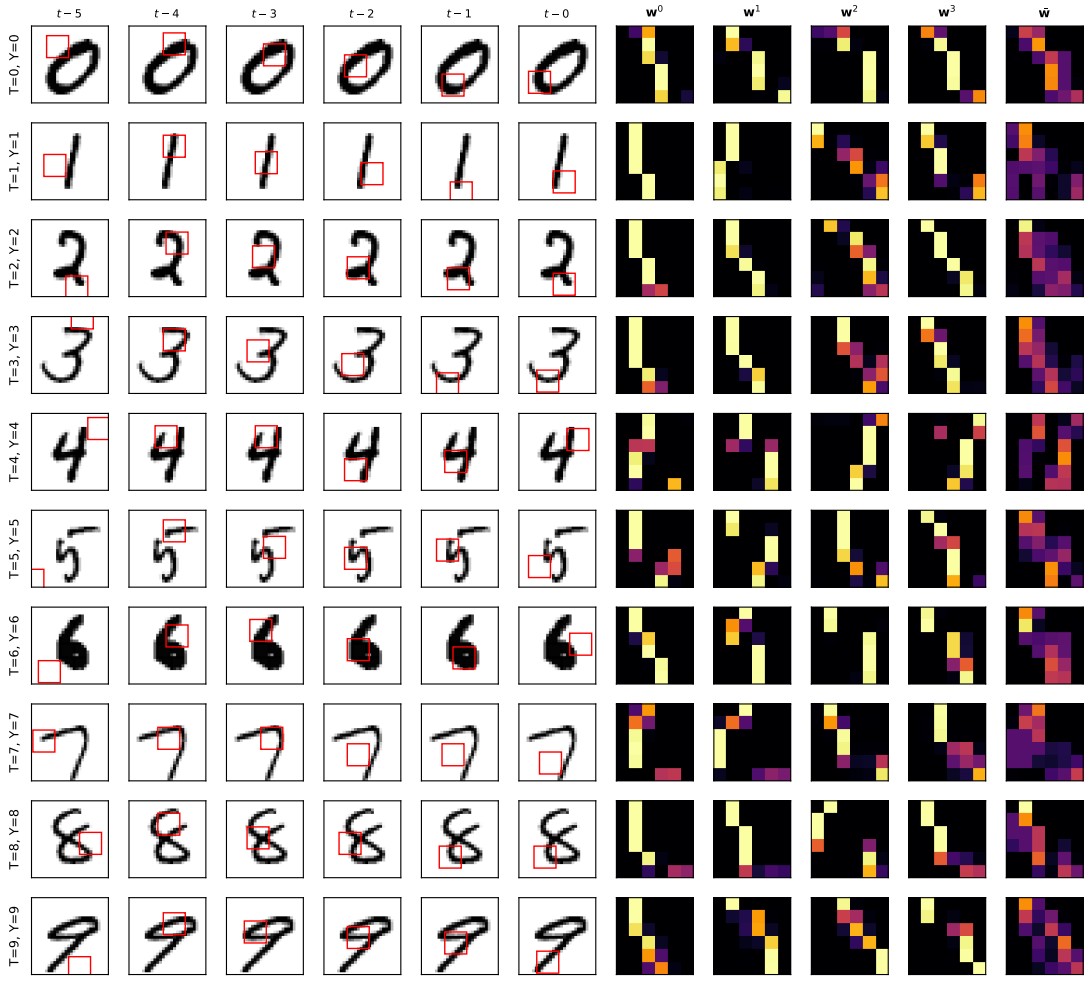

Figure 6: Example trajectories and distribution of attention weights for our best MNIST model.

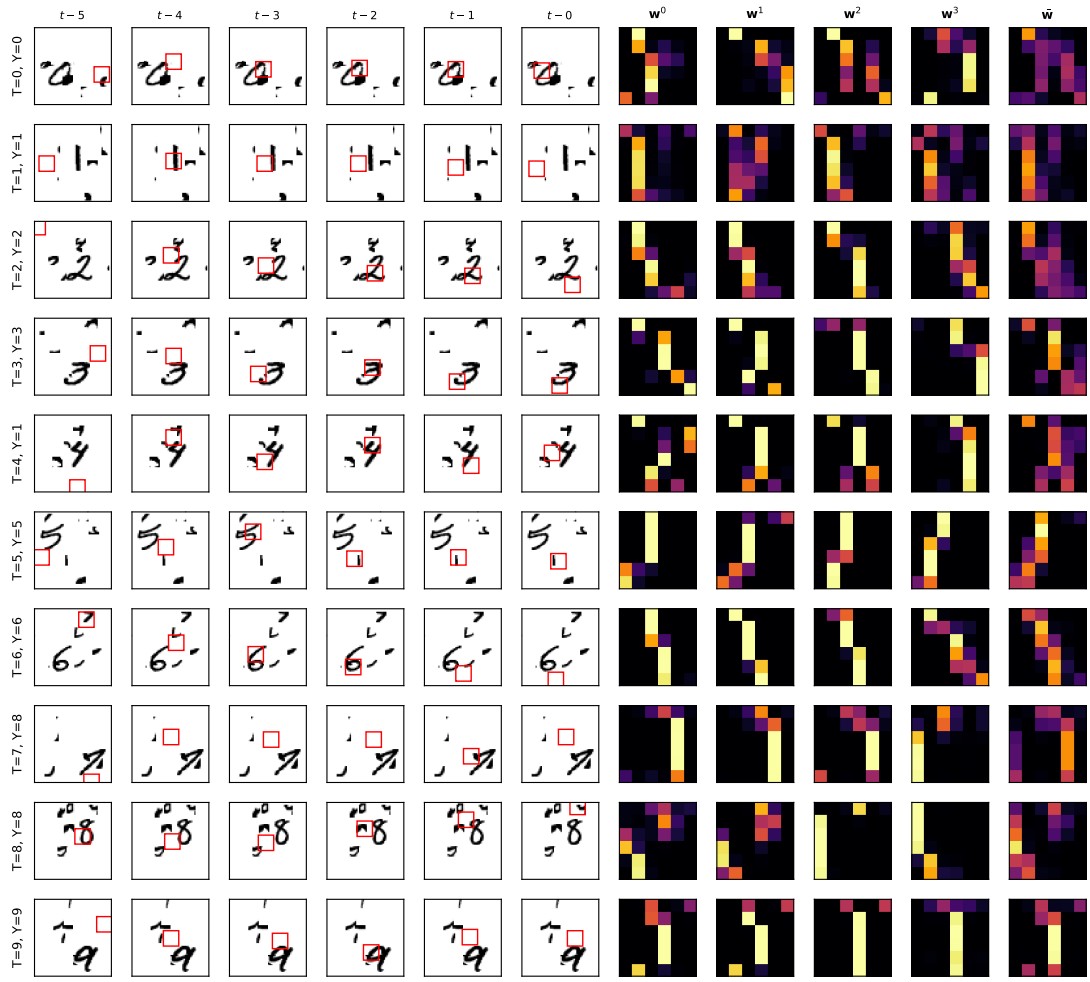

Figure 7: Example trajectories and distribution of attention weights for our best cluttered MNIST model.

## Appendix B. Visualizing Input Units

The glimpse network learns the "what" features from a glimpse of the input image. A fully-connected layer takes as input a glimpse before combining it with the "where" features. Our best model on the MNIST data uses 256 input units to represent this fully-connected layer. After we have trained the network, and made updates to the weights in this layer, we can visualize each unit separately. These visualizations are made to better understand, to some degree, how the network represents a glimpse and arrives at its prediction.

In no particular order, we show these units in Figure 8. These units operate on the unstandardized glimpse of MNIST digits with intensity values between $[0, 1]$. The majority of weights are positive (as shown in red), but reveal some interesting patterns. That is, there are strong positive gradients that outline the shape of certain lines and curves at

different rotations and angles. It is unclear why strong negative weights are in some of the corners. We speculate these could be to better inform the location network and it would be interesting to view the correlation of neuron activations and change in location values.

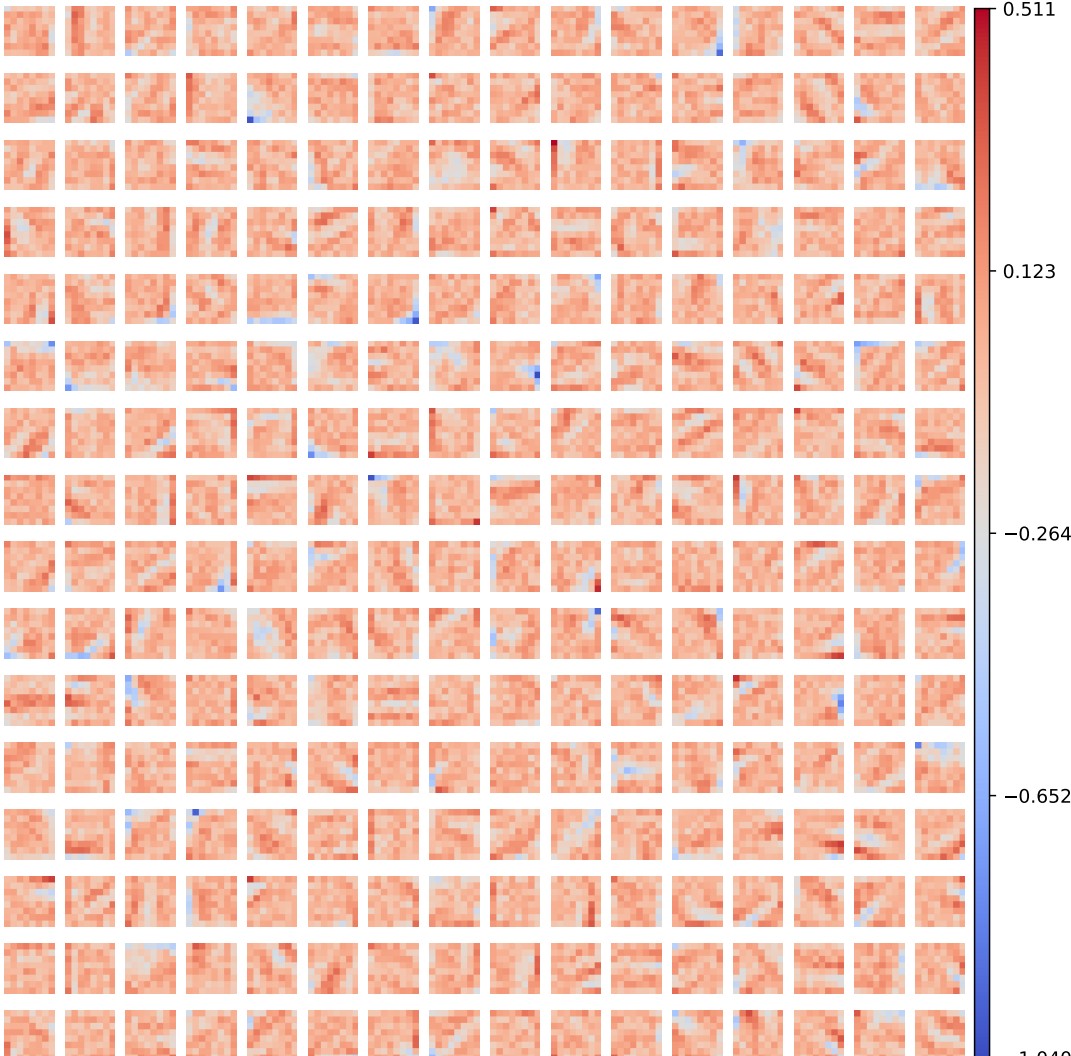

Figure 8: Input unit weights for the fully-connected layer in the glimpse network that takes as input an $8 \times 8$ glimpse of MNIST digits. In red are higher, positive weight values and blue are smaller and negative.

