# OpenReview forum: "Memory-Based Sequential Attention"
_NeurIPS.cc/2023/Workshop/Gaze_Meets_ML — Gaze Meets ML 2023 Oral_

### Official Review · Reviewer_2XE8 · 2023-10-21
**This is a high quality and well written paper. The method sounds and the results are interesting.**

**Rating:** 9
**Confidence:** 4

**Review:**

The paper is well-written. The bio-inspired framework that optimizes scan paths for image classification is interesting. The results show that the proposed framework can provide some explainability of the classification results. The use of a transformer-based memory module with reinforcement learning is also reasonable. More specific comments:

1. Why only a single transformer block was used? Did the ViT also have one block also? More blocks may improve the performance.
2. What are the units in Table 1 and 2? Percentage?
3. It can be interesting to see the results on nature images such as ImageNet or CIFAR-10.

---

### Official Review · Reviewer_KCGz · 2023-10-22
**Biologically motivated attention for better interpretability**

**Rating:** 6
**Confidence:** 5

**Review:**

The proposed work is closely related to two foundational papers: RAM and Dosovitskiy et al. Nevertheless, the authors introduce a novel hybrid approach that combines reinforcement learning and data likelihood estimation to determine an optimal trajectory over a visual scene. Additionally, their inspiration is drawn from a biological process associated with attention functioning. Importantly, the authors support their choice by referencing studies in psychology and neuroscience, providing clearer insights into the motivation behind their approach.

A fair and comprehensive comparison is made between the proposed approach and the closely related ideas of RAM and Dosovitskiy, emphasizing the significance of their contribution.

However, the authors present their work with the aim of achieving two primary objectives: enhancing overall performance and improving interpretability. Nevertheless, the question of achieving superior performance, especially when compared to the RAM approach (taking into account hyperparameter tuning), remains debatable. Therefore, it may be advisable to shift the main focus and emphasis of the paper toward interpretability.

Additionally, it is essential to clearly define and state the specific contributions of their work since certain components appear to be leveraged from either RAM or Dosovitskiy. This clarity will enhance the understanding of the novelty proposed by the authors.

---

### Official Review · Reviewer_RiP6 · 2023-10-25
**Review of Submission #7**

**Rating:** 7
**Confidence:** 3

**Review:**

SUMMARY: This work proposes a framework that couple transformer-based memory module coupled with a reinforcement learning-based learning algorithm for image classification to uncover biologically plausible scan paths. They empirically evaluate their approach on the MNIST and cluttered MNIST dataset to compare against vision transformers, CNNs and a recurrent model of visual attention. They also study the differences in visual attention trajectories using different initial fixations.

STRENGHTS: The methodology to emulate sequential attention using vision transformers and reinforcement learning for gaze sampling (glimpses) is very interesting. The experimental design on the two datasets is sound and tests important aspects of the work and presents interesting results.

QUESTIONS/WEAKNESSES:

1. The authors have experimented with MNIST variants primarily, which are relatively less complex to fit image datasets than natural images (where many interesting things may be present in the same image). It would be interesting to see how the model performs relative to ViTs and CNNs in this comparison

2. Perhaps I am missing something, but it is not clear whether the model can be made robust to different input resolutions of the images.

3. The experiments use a single data split, and thus do not quantify confidence thresholds for the error estimates. This raises questions on the robustness of their findings. Additionally, the only metric reported is classification error.

---

### Meta-Review · Area_Chair_Z6ju · 2023-10-25

**Recommendation:** Accept (Oral)
**Confidence:** 5

**Metareview:**

The authors have proposed a unique approach that combines the transformed-based approach with a reinforcement learning framework for image classification to determine the optimal trajectory over visual scenes.

The reviewers appreciated the paper and praised the quality of the writing, the comparison with existing approaches, and the innovative combination of reinforcement learning and vision transfers. However, they raised some questions about the results, mainly regarding the choice and simplicity of MNIST and the similarity to the RAM and ViT-based approaches. Nonetheless, the reviewers believe this paper would make an excellent workshop paper and lead to exciting discussions.

---

### Decision · Program_Chairs · 2023-10-26

Accept (Oral)